# Natural language processing to evaluate texting conversations between patients and healthcare providers during COVID-19 Home-Based Care in Rwanda at scale

**Richard T. Lester** [1]☯*, **Matthew Manson** [1]☯, **Muhammed Semakula**[2,3], **Hyeju Jang**[4,5], **Hassan Mugabo**[3], **Ali Magzari**[6], **Junhong Ma Blackmer**[7], **Fanan Fattah**[1], **Simon Pierre Niyonsenga**[3], **Edson Rwagasore**[3], **Charles Ruranga**[8], **Eric Remera**[3], **Jean Claude S. Ngabonziza**[3,9], **Giuseppe Carenini**[5], **Sabin Nsanzimana**[2]

1 Division of Infectious Diseases, Department of Medicine, University of British Columbia, Vancouver, British Columbia, Canada, 2 Rwanda Ministry of Health, Kigali, Rwanda, 3 Rwanda Biomedical Centre, Kigali, Rwanda, 4 Luddy School of Informatics, Computing, and Engineering, Department of Computer Science Indiana University Indianapolis, Indianapolis, Indiana, United States, 5 Department of Computer Science, Faculty of Science, University of British Columbia, Vancouver, British Columbia, Canada, 6 Department of Electrical and Computer Engineering, University of British Columbia, Vancouver, British Columbia, Canada, 7 Department of Mathematics, University of British Columbia, Vancouver, British Columbia, Canada, 8 African Center of Excellence in Data Science, University of Rwanda, Kigali, Rwanda, 9 Department of Clinical Biology, University of Rwanda, Kigali, Rwanda

☯ These authors contributed equally to this work.
* rlester@mail.ubc.ca

## Abstract

Community isolation of patients with communicable infectious diseases limits spread of pathogens but our understanding of isolated patients' needs and challenges is incomplete. Rwanda deployed a digital health service nationally to assist public health clinicians to remotely monitor and support SARS-CoV-2 cases via their mobile phones using daily interactive short message service (SMS) check-ins. We aimed to assess the texting patterns and communicated topics to better understand patient experiences. We extracted data on all COVID-19 cases and exposed contacts who were enrolled in the WelTel text messaging program between March 18, 2020, and March 31, 2022, and linked demographic and clinical data from the national COVID-19 registry. A sample of the text conversation corpus was English-translated and labeled with topics of interest defined by medical experts. Multiple natural language processing (NLP) topic classification models were trained and compared using F1 scores. Best performing models were applied to classify unlabeled conversations. Total 33,081 isolated patients (mean age 33·9, range 0–100), 44% female, including 30,398 cases and 2,683 contacts) were registered in WelTel. Registered patients generated 12,119 interactive text conversations in Kinyarwanda (n = 8,183, 67%), English (n = 3,069, 25%) and other languages. Sufficiently trained large language models (LLMs) were unavailable for Kinyarwanda. Traditional machine learning (ML) models outperformed fine-tuned transformer architecture language models on the native untranslated language corpus, however, the reverse was observed of models trained on English-only data. The most frequently identified topics discussed included symptoms (69%), diagnostics (38%), social issues (19%),

**Data Availability Statement:** The data from the COVID-19 DHIS2 and WelTel databases contain human subject data that may be sensitive. Although clinical record data were deidentified by removal of names, and the conversational data captured in WelTel by removing proper names, given the potential to identify human subjects through contextual information from people who know them, the data was not published in a public repository. Requests for access to raw data can be made to the Research, Innovation and Data Science Division at the Rwanda Biomedical Center, Kigali, Rwanda info@rbc.gov.rw.

**Funding:** The research was funded by the Canadian 2019 Novel Coronavirus (COVID-19) Rapid Research Funding Opportunity partnership (440231 to RTL) between Canadian Institutes of Health Research (CIHR), Natural Sciences and Engineering Research Council of Canada (NSERC), Social Sciences and Humanities Research Council (SSHRC), Canada Research Coordinating Committee (CRCC) through the New Frontiers in Research Fund (NFRF), International Development Research Centre (IDRC) and Genome Canada (GC). The WelTel project was funded by Grand Challenges Canada (TTS-2003-37605 to RTL) and matched in-kind support from the Rwanda Biomedical Center. The funders had no role in study design, data collection and analysis, decision to publish, or preparation of the manuscript.

**Competing interests:** RTL is co-founder and chief scientific officer of WelTel Incorporated. WelTel Incorporated received a grant from Grand Challenges Canada to provide the software and technical support in Rwanda during the pandemic. No other authors have conflict of interests to declare. Data stewardship and reporting was overseen by the Rwanda Biomedical Center.

prevention (18%), healthcare logistics (16%), and treatment (8·5%). Education, advice, and triage on these topics were provided to patients. Interactive text messaging can be used to remotely support isolated patients in pandemics at scale. NLP can help evaluate the medical and social factors that affect isolated patients which could ultimately inform precision public health responses to future pandemics.

## Author summary

We present the first application of NLP for categorizing text messages between patients and healthcare providers within a nationally scaled digital healthcare program. This study provides unique insights into the circumstances of home-based COVID-19 patients during the pandemic. Our trained topic classification models accurately categorized topics in both English and African language texts. Patients reported and discussed both medical and social issues with public healthcare providers. This approach has the potential to guide precision public health decisions and responses in future outbreaks, pandemics, and remote healthcare scenarios.

## Introduction

The practice of **isolating patients with communicable infectious diseases** reduces transmission by separating infected individuals from healthy ones, and isolation at home allows healthcare systems to focus on managing severe cases in hospitals [1]. However, isolated patients may have changes to their condition that require attention from medical professionals and can experience additional social stressors [2]. Isolation protocols have even been considered a risk factor for mortality [3]. Understanding the complexity of facility and home-based isolation remains a critical consideration in effective pandemic response planning.

At the pandemic outset, the Rwanda government COVID-19 Joint Task Force adopted a short message service (SMS)-based digital health service, WelTel, to remotely monitor and support community isolated cases and contacts of SARS-CoV-2 from the centralized national command post. Its use continued throughout the emergency phase of the pandemic as the key to their Home-Based Care (HBC) program [4]. The WelTel platform was selected based on its history of use in HIV programs, prior clinic evidence, and importantly, its accessibility [5–7]. Internet based tools and smartphone apps would have been insufficient since, as a lower-middle income country (LMIC), only 23% of the Rwanda population has internet-access; yet at least 90% of the population has access to cellular phones [8,9]. As in other applications of the WelTel platform, throughout the COVID-19 response in Rwanda, automated 'check-in' messages were sent daily to isolated patients, to which they could respond to report on their status and ask questions, presenting a novel data source for evaluating their needs and experiences. A manually performed topic classification of WelTel text messaging conversations with patients during HIV care previously demonstrated that these open-ended text messaging conversations contain wide ranging issues experienced by patients that could be acted upon. There is limited data on the challenges facing patients being monitored in an acute rapidly spreading pandemic such as the case with the recent COVID-19 pandemic [10].

Advances in natural language processing (NLP) enables rapid computational topic analysis (classification and modelling) which can delineate large amounts of text into topics of interest [11]. For example, topic classification models using machine-learning (ML) have been used in

clinical settings to label text messages exchanged between healthcare providers and outpatients [12–15]. However, such studies have often focused on a narrow list of topics or have been limited to demonstration projects or a single healthcare organization setting. Furthermore, there is an imbalance in healthcare data used to train and develop NLP models that threatens to leave entire continents behind in the development of artificial intelligence (AI) for healthcare [16].

In this study, we aimed to analyze the complete corpus of SMS conversations between Rwanda's public health clinicians and isolated COVID-19 patients and contacts during the first two years of the COVID-19 pandemic to get a better understanding of the experiences and needs of COVID-19 cases and contacts under community isolation. We considered a clinical, social, and logistical perspective using topic classification to better delineate the needs and experiences of patients and the advice they were given, during isolation–a large-scale approach that could be used to guide decision-making in future outbreak responses.

## Methods

### Setting and digital health monitoring technology

Rwanda, a Central African nation with a population of 14 million, was internationally recognized for its prompt response to the COVID-19 pandemic [17]. Their public health command implemented the WelTel text messaging service [www.weltelhealth.com] on March 18, 2020, coincident with detecting its first COVID-19 cases. This tool enabled public health staff to remotely monitor and support individuals who tested positive for, or were at high risk of contracting, SARS-CoV-2, and remained operational until the public health emergency was officially declared over.

The WelTel platform is a secure web-based application that healthcare providers can access from any internet-connected device. It interfaces with cellular networks enabling it to send both pre-set and manually input SMS messages to registered mobile phone users. It features a conversation dashboard that visualizes responses.

Following the initial facility-based quarantine of COVID-19 cases identified at border entry points, Rwanda launched its HBC program when community transmission became widespread. Under this program, mild cases and contacts were asked to stay home and be remotely monitored. Patients with significantly worsening clinical status were triaged to hospitals. Individuals were enrolled as 'cases' if they tested positive for SARS-CoV-2 or as 'contacts' if they were deemed high risk due to exposure. Patients without personal cell phone numbers were registered via a household member, friend, or neighbor.

The messaging protocol for COVID-19 monitoring involved sending bulk automated daily 'check-in' texts in Rwanda's principal languages (Kinyarwanda, English, French) daily to registered patients throughout their isolation period. Patients could respond via text in their own words to indicate their status and/or ask questions. Replies indicating a problem or question were flagged for follow-up. Public health clinicians responded to flagged messages to provide information or advice, forming interactive text conversations.

### Ethics approval

The University of British Columbia Clinical Research Ethics Board (CREB) and the Research Ethics Board of the Rwanda Biomedical Centre (RBC) approved this study.

### Included datasets and description

Patient registration and all text communication data generated between March 18, 2020 and March 31, 2022 was exported from the WelTel database and linked to the Rwanda

DHIS2-based COVID-19 testing registry to obtain missing demographic or clinical data [18]. Personal identifiers were removed. For the purposes of our study, a 'conversation' was defined as a text exchange consisting of at least one 'incoming' patient message containing three or more words (filtering out minimal replies e.g., 'doing well'), together with at least one 'outgoing' public health clinician or automated system message (thus being interactive). Patient registration and numbers of conversations and their characteristics (# messages, language used) were quantified. Conversation usage was compared across patient sociodemographic and clinical characteristics (sex, age, province of residence, pandemic wave, and COVID-19 status (case vs. contact)) using a multivariable logistic regression model. Odds ratios (OR) are reported with 95% confidence intervals (CI) and $p < 0.05$ was considered significant. Example conversations were selected to illustrate context.

## Language translation and topic labelling

A list of topics and subtopics was developed through a three-round mini-Delphi process involving ten public health experts, clinicians, and researchers from Rwanda and Canada. This list was an expansion of a previously published list focused on clinical care and used to develop a conversation analysis and visualization tool, ConVIScope [12]. The final list of topics and subtopics encompassed perspectives on public health, emergency response, and clinical care. Topic definitions are provided in (S1 Fig).

For the purpose of labeling a training dataset for topic classification models, a continuous sample of 2,791 text conversations from March 18, 2020, to May 31, 2021, was extracted from the study period corpus during the initial phases of the pandemic in order to start translation to English and begin the labeling process while the pandemic was ongoing and data continued to accrue. These conversations were analyzed for language use with Google's CLD2 language detector. Non-English sample messages were translated into English by a compensated professional healthcare text translator proficient in Kinyarwanda, English, French, and Kiswahili.

The English-translated conversations were labeled by at least three trained members of the research team using a custom conversation annotation tool. Any conflicts in labeling were resolved by an external team member. The topic labels were then applied back to the corresponding conversations in their original language, where applicable.

## Classification model training, testing, and application

We aimed to identify the best performing classification model for each topic and subtopic. Training set sample size constraints limited our classification experiments to topics with $\geq 100$ occurrences. We employed the binary relevance strategy, which simplifies performance management by building a separate binary classifier for each topic. We experimented with both traditional ML algorithms and Transformer-based language models for each topic. We explored large language models (LLMs, e.g., GPT-4), however, our literature review revealed that, of the three pre-trained with any Kinyarwanda, the pre-training datasets for these models contained minimal Kinyarwanda text ($\leq 0.01\%$). Investigations of the performance of these LLMs at NLP tasks in Kinyarwanda suggested they were insufficient to include [19].

*Traditional-ML classification models*: For training traditional ML models, we explored five feature sets: binary and non-binary bag of words, TF-IDF (term-frequency inverse document frequency), and character n-gram (with/without TF-IDF). The character n-gram features were tried because Kinyarwanda is an agglutinative language. All combinations of one of these feature sets, and one classification algorithm, were tested. We evaluated logistic regression, ridge classifier, and random forest as classification algorithms. Hyperparameters for each classifier were tuned using random search in five-fold cross validation [20].

*Transformer-Based Language Models*: We leveraged open-source pre-trained Transformer-based language models for Kinyarwanda, including AfriBERTa, AfroLM, AfroXLMR, and Kinya-BERT [21–24]. To fine-tune these models for our task, we employed the AdamW optimizer along with the binary cross-entropy loss function. Additionally, we incorporated dropout layers (with a probability of 0.0 for the first layer and 0.3 for subsequent layers) for regularization.

We compared the performance of each model for each topic using the validation set F1-score, with a minimum F1-score cut-off of 0.7. The highest F1-score models were applied to predict topics for the remaining 9,328 unlabeled, untranslated SMS conversations.

*Additional Language experiments*: Since our corpus contained a mix of high- and low-resource languages (e.g., English, French, Kinyarwanda) we explored how this might affect performance of the models by conducting identical experiments to above using only the English-translated version of the training corpus. We evaluated both traditional ML models (as above), and English pre-trained Transformer models (BERT, Longformer) [25,26]. F1 scores are reported for comparison.

See (S2 Fig.) for more hyperparameter and fine-tuning details.

## Topic analysis and statistics

Counts and proportions of conversations containing each topic (full corpus), and subtopic (sampled corpus), are provided. Topic frequencies were compared by patient demographics and clinical metrics using multiple independent multivariable logistic regression models for each topic. Models incorporate all patient demographic and clinical characteristics and use complete cases. OR are reported with 95% CIs and p-values are adjusted for multiple testing using the Benjamini-Hochberg correction with a false discovery rate of 0·05. Classified topic categories could be batch filtered and manually read to understand the nature of the discussions that occurred within each topic [12]. Data summarizations, visualizations, and statistical testing were produced or conducted using R version 4.2.1.

## Results

### Participation in text messaging conversations

During the study period, Rwanda reported 131,190 cases of COVID-19, of whom 33,081 individuals (25%, 30,398 cases, 2,683 contacts) were registered in the WelTel texting service. The numbers of patients registered, and the occurrence of text-message conversations (with at least 3 interactive messages), appeared in waves similar to the COVID-19 incidence reported in the national registry (Fig 1).

Of those registered, 6,021 patients (18%) used the texting service to generate 12,119 conversations (Table 1). Controlled for other demographic and clinical factors, the odds of generating a conversation among females compared to males was 0·83 (95%CI: 0·78–0·89; p<0·001). Age did not significantly influence the frequency of conversations (OR per year increase in age = 1·00, 95%CI: 0·99–1·00, p = 0·17). Contacts were more likely to generate conversations than cases (OR 1·83 95%CI: 1·63–2·06, p<0·001). The proportion registered patients texting conversations varied across waves of the pandemic (Wave one: 12%; two: 37%; three: 25%; and four: 3%).

The median number of conversations per patient was one (IQR: 1–2, range: 1–18), and the median conversation length (# of discrete text messages) was three (IQR: 2–5, range: 2–44). Most conversations occurred in Kinyarwanda (8,183, 67%) or English (3,069, 25%), however other regional languages (French, Kiswahili, Ganda, Chewa/Nyanja) were also identified (<10%). Language proportions were similar to the proportion of the population considered literate in those languages in Rwanda [27].

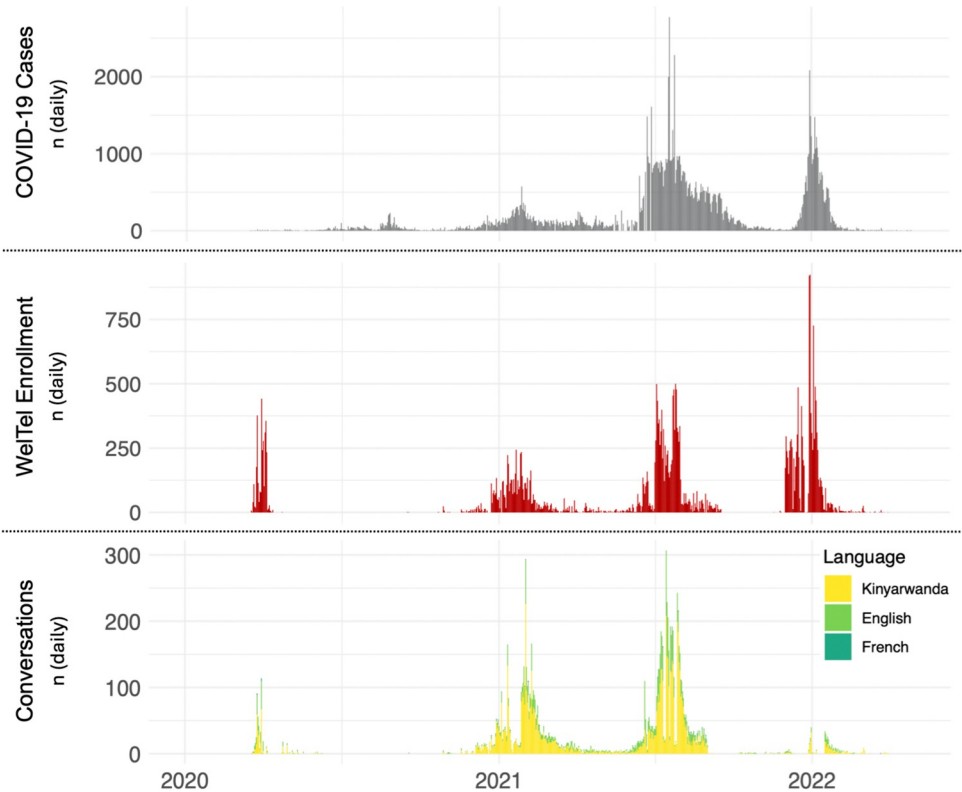

**Fig 1. COVID-19 case incidence, and enrollment and use of the WelTel SMS text messaging service during the COVID-19 pandemic in Rwanda.** The grey plot captures the four major waves of case incidence in Rwanda. The red plot captures enrollment of COVID-19 patients (cases and contacts) in the WelTel system throughout the study period. Between June and September 2020, patients were not being enrolled in WelTel as the Rwandan government transitioned to government servers to host the software. The yellow and green plot captures the number of patient-provider SMS conversations throughout the study period; most conversations occurred in Kinyarwanda (67%) and English (25%) and other regional languages.

## Comparison of NLP model performance

The NLP models tested classified most topics with a level of performance above our cut-off F1-score (Table 2). The traditional ML models performed better than fine-tuned transformer models on untranslated native language conversation data. Best performing models for each topic met the pre-determined F1-score cutoff of 0·7 for six of the nine topics meeting the sample size threshold of 100. Model performance was roughly correlated with the number of occurrences (within sample) of the given topic.

In our language experiments, fine-tuned transformer models generally performed better when using the English (translated) training and testing corpus, exceeding the F1 threshold in all but one topic.

## Topic analysis of conversations

Human labeling (training corpus) and machine topic labeling (full corpus) identified the topics and subtopics of interest at roughly similar frequencies (Fig 2 and S3 Fig). Overall, medical topics were discussed more frequently than non-medical topics. The most frequent topic was symptoms (69%), the majority reporting a lack of symptoms (59%). Diagnostic methods (38%, e.g., receiving COVID-19 test results), prevention (18%, including non-pharmaceutical such

**Table 1. Characteristics of COVID-19 patients (cases and contacts) and usage of the WelTel text messaging service in Rwanda between March 18, 2020, and March 31, 2022.** *Percentages are within group. A proxy is an individual who allowed another/other individual(s) (e.g., a non-phone owner) to register for WelTel using their phone number (e.g., a mother could be a proxy for her non-phone-owning daughter). Shared refers to the individuals using a proxy (e.g., the daughter).*

| | People | | | Conversations | | |
|---|---|---|---|---|---|---|
| | COVID-19 CASES | ENROLLED | PRODUCING CONVERSATIONS | CONVERSATION LENGTH | | |
| | | | | Median # Messages | IQR | Range |
| **TOTAL** | **131,190** | **33,081** | **6,021** | 3 | 2–5 | 2–44 |
| **COVID-19 Status (at registration)** | | | | | | |
| Case | 131,190 (100%) | 30,398 (92%) | 5,322 (88%) | 3 | 2–5 | 2–44 |
| Contact | - | 2,683 (8%) | 699 (12%) | 4 | 3–8 | 2–30 |
| **Location of Residence** | | | | | | |
| Kigali City | 42,152 (32%) | 33,018 (97%) | 5855 (97%) | 3 | 2–5 | 2–44 |
| Northern | 20,347 (16%) | 111 (0.3%) | 22 (0.4%) | 4 | 3–6 | 2–16 |
| Southern | 25,814 (20%) | 370 (1%) | 60 (1%) | 3 | 2–5 | 2–18 |
| Eastern | 18,007 (14%) | 233 (1%) | 25 (0.4%) | 5 | 3–7·5 | 2–11 |
| Western | 24,870 (19%) | 161 (0.5%) | 15 (0.2%) | 3 | 2–4·5 | 2–16 |
| Unknown | - | 188 (0.6%) | 44 (0.7%) | 4 | 2–9 | 2–19 |
| **Sex** | | | | | | |
| Male | 63,637 (49%) | 17,102 (52%) | 3,385 (56%) | 3 | 2–5 | 2–36 |
| Female | 67,545 (51%) | 14,665 (44%) | 2,608 (43%) | 3 | 3–5 | 2–44 |
| Unknown | 8 (0.0006%) | 1,314 (4%) | 28 (0.5%) | 4 | 3–7 | 2–14 |
| **Age** (mean (SD, range)) | 36·2 (17·36, 0–115) | 33·9 (14·51, 0–100) | 34·0 (13·27, 0–91) | - | - | - |
| **Proxy/shared** | - | 1,726/2,749 | 453 (-) | - | - | - |

as how to maintain isolation, and pharmaceutical such as vaccines), healthcare logistics (16%, e.g., asking where to get tested), and treatment (8%, including management of symptoms). Social topics (19%) included mention of culture or religion (5·5%), followed by discussing friends and family (4·9%). Manually labeled training data revealed important issues such as nutrition and food security (3·7%). Examples of text conversations are provided in Table 3.

Comparing the odds of topic discussion within the full corpus by patient demographic and clinical factors revealed minor differences that are reported in (S1 Table).

## Discussion

To our knowledge, this is the first study to report on the use of NLP to investigate a broad range of topics from a complete corpus of conversational texts with patients in home isolation in a pandemic or outbreak setting at large scale. Through a combination of manual annotation and NLP techniques, we gained insights into patient experiences during COVID-19 in Rwanda by identifying key discussion topics between patients and public health clinicians. The sustained use of the text messaging service in Rwanda demonstrated the feasibility of using SMS-based communication for remote monitoring at scale in a region with limited internet access but widespread mobile phone penetration. Leveraging existing infrastructure allowed for efficient support for isolated individuals, regardless of location or access to healthcare facilities.

Our data captured the first two years of the Rwandan COVID-19 pandemic, including text conversations in Kinyarwanda, English, and other local languages with public health clinicians to report their status or seek advice. Topic classification revealed that medical topics, such as symptoms, diagnostics, prevention, and treatment, were commonly discussed, reflecting patients' primary focus on their health status and obtaining medical guidance. However, social and lifestyle topics, including discussions about social support, healthcare logistics, diet/

**Table 2. Performance of classification models on original language (untranslated), and English-translated conversations for each topic of interest.** *Blue and orange fill indicate traditional-ML, and transformer architecture language models, respectively. Bold text indicates an F1 score meeting our performance cutoff (≥0.7). Topics are ordered by best-performing model F1 performance score (descending) when trained with original language (untranslated) conversations as ultimately used to label the full corpus.*

| Topic (n) | Untranslated Conversations | | English-translated Conversations | |
|---|---|---|---|---|
| | Classifier (Feature Extraction Method) | Performance (Precision Recall F1) | Classifier (Feature Extraction Method) | Performance (Precision Recall F1) |
| **Diagnostic Methods** (1030) | **Random Forest** (Character n-gram + TF-IDF) | 0.93 0.90 **0.92** | **BERT** (NA) | 0.95 0.97 **0.96** |
| **Symptoms** (2002) | **KinyaBERT** (NA) | 0.87 0.94 **0.91** | **BERT** (NA) | 0.92 0.96 **0.94** |
| **Treatment** (174) | **KinyaBERT** (NA) | 0.85 0.88 **0.90** | **Longformer** (NA) | 0.96 0.96 **0.96** |
| **Prevention** (459) | **Ridge** (Character n-gram + TF-IDF) | 0.77 0.84 **0.81** | **Longformer** (NA) | 0.97 0.93 **0.95** |
| **Social** (406) | **Logistic Regression** (Character n-gram + TF-IDF) | 0.67 0.74 **0.70** | **Longformer** (NA) | 0.88 0.82 **0.85** |
| **Healthcare Logistics** (281) | **Logistic Regression** (TF-IDF) | 0.62 0.81 **0.70** | **Ridge** (Character n-gram + TF-IDF | 0.64 0.88 **0.74** |
| **Service Quality** (425) | **Ridge** (Character n-gram) | 0.69 0.67 0.68 | **Longformer** (NA) | 0.83 0.77 **0.80** |
| **Lifestyle/ Behavioural** (156) | **Logistic Regression** (TF-IDF) | 0.71 0.65 0.68 | **Longformer** (NA) | 0.92 1.00 **0.96** |
| **Technical/ IT** (185) | **Logistic Regression** (Non-Binary Bag of Words) | 0.80 0.43 0.56 | **BERT** (NA) | 0.83 0.54 0.65 |

nutrition, finances, and service quality, also emerged as important aspects to patients during isolation. The data underscores the need for holistic support systems for patient isolation programs. Additionally, while our topic classification approach rapidly quantified prevalent topics well, reflecting their relative importance to patients, it also enables the sorting of large datasets into smaller categories for easier manual evaluation by decision-makers. Topics such as food insecurity, even if less common than topics such as reporting symptoms, may still provide actionable targets for public health responders, especially if it can be localized to communities.

NLP techniques are rapidly being introduced into health systems to understand unstructured data from large datasets, such as clinical notes in electronic medical records (EMRs), social media posts, or free-text survey responses [11]. For example, topic classification has been employed to sort unstructured patient feedback from text messages [15]. Still, manual qualitative analyses dominate. A manually labeled thematic analysis of 1,454 patient portal secure messaging conversations between cancer patients and health professionals in the US during COVID-19 found 26% of conversations were related to COVID-19 and identified topics related to changes in care plans, symptoms, risks, precautions and symptoms [28]. In Northern Thailand manual topic classification of SMS exchanges with 213 home-isolated patients with diagnosed COVID-19 revealed several complexities of care including discussing

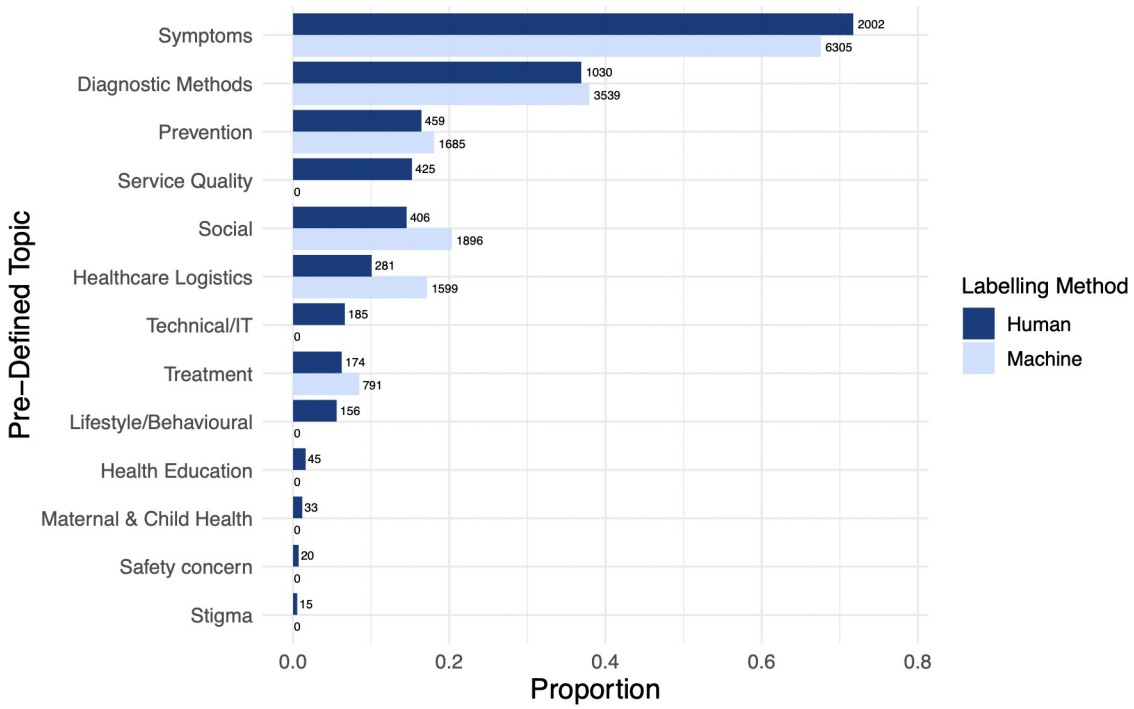

**Fig 2. Proportion of conversations containing each topic of interest within the complete corpus (n = 12,119) based on both human- (n = 2,791) and machine-labelling (n = 9,328).** Note: conversations were often labelled with multiple topics/subtopics therefore proportions do not sum to 1. Some topics did not have sufficient data to produce models, or the models developed did not meet our performance cutoff of F1≥0.7. For these, the proportion estimates are based upon human labelling alone.

symptoms or vital signs, medications, diet and exercise, mental stresses, information about COVID-19, and a high proportion of logistical assistance to navigate the health services [29]. Manual labeling is resource intensive and generally limited to smaller data sets. One US-based study used NLP techniques to determine if patient-initiated secure messaging via a EMR portal was associated with positive testing, but this was a simple binary classification [13]. To our knowledge, our study is the first to use NLP techniques to classify patient clinician texts into broad medical, social and logistical topics set by experts, and at a national scale.

Our study compared performance of several ML model approaches. Traditional ML models and transformer-based language models were both effective in classifying text message conversations into the predefined topics. While traditional ML models performed better with untranslated conversations, transformer models had superior performance when trained on English (translated) data, highlighting their limited performance on African languages [30]. This observation is consistent with the knowledge that transformer architecture language models are likely to underperform on NLP tasks in low-resource languages until sufficient high-quality pre-training data is sourced and leveraged [31]. Similarly, LLMs, which are also based on transformer architecture, currently contain too little, if any, training on Kinyarwanda and other low-resource languages to be useful in our context [19]. A brief trial using Google Translate to machine translate Kinyarwanda text into English for our study was inaccurate. This highlights the importance of considering language-specific model performance when analyzing multilingual text data [32]. Language consideration remains a significant challenge, with over 4,000 written languages worldwide. Pandemics affect everyone and AI tools must account low-resource languages to be equitable.

**Table 3. Illustrative text conversations excerpts between public health officials and patients (cases and contacts) during the COVID-19 pandemic in Rwanda.** *"..." indicates excluded text for brevity. Message types include: '(S):'–a 'system' message containing customizable content with automated sending; '(P):'–a 'patient' message containing patient remarks in free text form; and '(C):'–a 'clinician' message containing clinician remarks in free text form.*

| CONVERSATION TOPIC | EXAMPLE |
|---|---|
| Reporting a lack of symptoms | *(S): Hello, how are you today? Do you have any symptoms of coronavirus? [alongside Kinyarwanda and French translations of the message]*<br>**(P): We are ok. We do not have symptoms.**<br>(C): Hello! We inform you that you were tested negative for cornavirus. Thank you. |
| Reporting symptoms | *(S): Are you OK (Yes/No)? Or tell us if you have new symptom [alongside Kinyarwanda translation of the message]*<br>**(P): Good morning. I am coughing and I have flu but my situation is not alarming. I do not think that there is a problem.**<br>(C): Thank you for informing us. Explain your situation to the Doctor who is in charge of calling you. He is going to call you soon. |
| Diagnosis | *(S): haven't heard from you yet. how are you? [alongside Kinyarwanda translation of the message]*<br>**(P): Good morning. My name is #NAME#. I have been suffering from COVID-19 for days, but yesterday I went at #HEALTH FACILITY# to have a medical test for checking. Till now I have not yet got results via an sms. Can you help me to get those results if possible?**<br>(C): Good morning. We will inform people in charge to help you. Be patient.<br>(C): #NAME# please note that you tested Negative on COVID-19 Ag RT on 2021-01-25. Code: ********* |
| Treatment | *(S): Check-in message*<br>**(P): "...I took a Doliprane. Because I have little fever. 37.4..."**<br>(C): "Nice to hear that you are doing well.Doliprane helps for fever, body aches and headache..."<br>**(P): Yes, Thank you! Have a nice day** |
| Pharmaceutical prevention | *(S): Check-in message*<br>**(P): "...Can you tell me where people suffering from heart issues and other diseases can go to get a vaccine?..."**<br>(C): "...Where do you usually go for a medical follow up...?"<br>**(P): "...#HEALTH FACILITY#"**<br>**(P): I suffered from Covid 19 a month ago but now I am healthy.**<br>(C): You can contact them in order to know what they have scheduled for you.<br>**(P): Thank you** |
| Non-pharmaceutical prevention | **(P): "...[Our] child has already tested positive...I have some questions...Will not I be infected again once I share the same room and equipments with my child?...Is it possible to get a medical test for checking before ending 14 days of quarantine?..."**<br>**(P): "...I am still suffering from coughing while the medicine I was taking is over..."**<br>(C): "...It is a must to go for a medical test for checking after 14 days. It would be better to clean your hands neatly, and to put on mask accordingly before taking your child..."<br>**(P): Thank you very much.** |
| Social–food and nutrition | **(P): "...I am suffering from COVID-19 and I live alone. I am starving and I don't have food. What can you help me? Thank you for your helpful answer."**<br>(C): Good afternoon. You can call the head of your village for a help. Food aid is being distributed these days. Thank you. |
| Social–culture and religion | *(S): Check-in message*<br>**(P): We are well this morning. We are really under the protection of Jesus.**<br>(C): That is good if you are well. Have a nice day. |
| Maternal & child health | *(S): Check-in message*<br>**(P): Hello. I am pregnant and I have tested positive for COVID-19. At the beginning I did not have any symptoms, but now I have many and I need your advice.**<br>(C): "...Be patient, you are going to get our support via a telephone call." |

*(Continued)*

**Table 3.** (Continued)

| CONVERSATION TOPIC | EXAMPLE |
|---|---|
| Gratitude | (C): Hello, do you have any health or COVID related concerns you would like to share with us today?<br>**(P): I tested negative for Covid 19. Thank you.**<br>**(P): Thank you for your regular care.**<br>**(P): Your advices are very important and we will always resort to them.** |
| Safety concern | (S): *Check-in message*<br>**(P): We are in a very bad situation. . . .We are mixed with people who have symptoms and those who do not have them, and we share the same bathroom and shower. Please, help us to get out of this situation before being infected.**<br>(C): Thank you for informing us. We are following up your situation. Keep on informing us. Thank you. |
| Proxy example | (S): *Check-in message*<br>**(P): Good morning. All of us, including children, are well. We don't have a problem. We hope to recover soon. Have a nice day. Thank you.**<br>(C): Thank you and have a nice day too. |

Our study has limitations. First, the supervised learning approach required the pre-definition and manual labeling of topics of interest which can be human resource intensive. Furthermore, the training data sample was collected from the initial phases of the pandemic, yet topics of interest or training samples annotated may evolve over time or change in different settings requiring continuously updating training datasets. For example, prevalent discussions of vaccines came later in the pandemic after most of our training data had been annotated. Including unsupervised ML models that discover topics as they emerge could help overcome this bias but have their own flaws. Furthermore, while our dataset was similar in size to those used to train other topic classification models in healthcare, many subtopics of interest were not prevalent enough to develop models. Additionally, the training dataset was translated into English for labelling purposes which could result in nuances of meaning being lost in translation. A professional translator fluent in the languages utilized and knowledgeable in public health was used and the topics of interest were fairly broad, making it less likely that such topics would be missed by slight interpretation variance of translations. The interpretation of the classified data in our study was done manually. However, if using AI to make interpretations or recommendations should continue to be done with caution and requires verification steps. Finally, training the models took time and conducting the NLP analyses afterwards meant the data was not available to decision-makers in real time. Ideally, once models are trained and validated, they could be applied and even updated using active machine learning while an outbreak or pandemic is occurring. Applying these models could then provide insights into patient and population issues that could be responded to while they are occurring and most relevant. Together with location data, this type of analysis could contribute to precision public health responses.

## Conclusion

Moving forward, future research should explore ways to further optimize the evaluation of text-based communication for greater insights into the patient care journey in pandemics, and clinical care. It must consider technology infrastructure available to populations globally and be inclusive of resource-limited languages [33,34]. Efforts to improve language support and translation capabilities across diverse linguistic and cultural contexts will be essential to ensure equitable access to digital services by people in all regions. This study demonstrated how large numbers of natural texting conversations between patients in the community and public

healthcare providers could be categorized into relevant medical and behavioral-social topics for rapid comparison of relative frequencies, or for easier review by such topics of interest by investigators or interested stakeholders once validated models are developed. Ultimately, emerging research on models that detect, understand, and generate natural language, as well as development of tools that formulate predictions and recommendations for healthcare decision-makers (generative AI) will be available and need to be validated to achieve the higher accuracy, precision, and reliability required for use in routine healthcare.

## Supporting information

**S1 Fig. Defined topics and subtopics of interest were determined by a Delphi process.**
(PDF)

**S2 Fig. NLP Model Details.**
(PDF)

**S3 Fig. Frequency and proportion of conversations in which all topics and subtopics of interest occurred within the human-labelled, machine-labelled, and complete corpora.**
(PDF)

**S1 Table. Comparison of conversation topics of interest by patient demographic and clinical factors.**
(PDF)

## Acknowledgments

We would like to thank the staff at the Rwanda Biomedical Center and work-learn students at the University of British Columbia who helped with language translation, annotation, and various aspects of background research and model building.

## Author Contributions

**Conceptualization:** Richard T. Lester, Muhammed Semakula, Hyeju Jang, Hassan Mugabo, Ali Magzari, Charles Ruranga, Giuseppe Carenini, Sabin Nsanzimana.

**Data curation:** Matthew Manson, Muhammed Semakula, Hyeju Jang, Hassan Mugabo, Ali Magzari, Junhong Ma Blackmer.

**Formal analysis:** Matthew Manson, Muhammed Semakula, Hyeju Jang, Hassan Mugabo, Ali Magzari, Junhong Ma Blackmer.

**Funding acquisition:** Richard T. Lester, Muhammed Semakula, Fanan Fattah, Sabin Nsanzimana.

**Investigation:** Richard T. Lester, Matthew Manson, Muhammed Semakula, Hyeju Jang, Hassan Mugabo, Ali Magzari, Junhong Ma Blackmer, Fanan Fattah, Edson Rwagasore, Charles Ruranga, Eric Remera, Jean Claude S. Ngabonziza, Giuseppe Carenini, Sabin Nsanzimana.

**Methodology:** Richard T. Lester, Matthew Manson, Muhammed Semakula, Hyeju Jang, Hassan Mugabo, Ali Magzari, Junhong Ma Blackmer, Charles Ruranga, Giuseppe Carenini.

**Project administration:** Richard T. Lester, Matthew Manson, Muhammed Semakula, Hyeju Jang, Hassan Mugabo, Ali Magzari, Fanan Fattah, Simon Pierre Niyonsenga, Edson Rwagasore, Eric Remera.

**Resources:** Richard T. Lester, Muhammed Semakula, Edson Rwagasore, Eric Remera, Sabin Nsanzimana.

**Software:** Hassan Mugabo, Ali Magzari.

**Supervision:** Richard T. Lester, Muhammed Semakula, Hyeju Jang, Ali Magzari, Simon Pierre Niyonsenga, Edson Rwagasore, Eric Remera, Jean Claude S. Ngabonziza, Giuseppe Carenini, Sabin Nsanzimana.

**Validation:** Muhammed Semakula, Hyeju Jang, Hassan Mugabo, Ali Magzari, Junhong Ma Blackmer, Fanan Fattah, Charles Ruranga.

**Visualization:** Matthew Manson, Muhammed Semakula, Hyeju Jang, Ali Magzari, Junhong Ma Blackmer.

**Writing – original draft:** Richard T. Lester, Matthew Manson, Muhammed Semakula, Ali Magzari.

**Writing – review & editing:** Richard T. Lester, Matthew Manson, Muhammed Semakula, Hyeju Jang, Hassan Mugabo, Ali Magzari, Junhong Ma Blackmer, Fanan Fattah, Simon Pierre Niyonsenga, Edson Rwagasore, Charles Ruranga, Eric Remera, Jean Claude S. Ngabonziza, Giuseppe Carenini, Sabin Nsanzimana.

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
