## [Decision Letter · Decision Letter 0]

15 Oct 2024

PDIG-D-24-00364

Natural language processing to evaluate texting conversations between patients and healthcare providers during COVID-19 Home-Based Care in Rwanda at scale

PLOS Digital Health

Dear Dr. Richard Todd Lester,

Thank you for submitting your manuscript to PLOS Digital Health. After careful consideration, we feel that it has merit but does not fully meet PLOS Digital Health's publication criteria as it currently stands. Therefore, we invite you to submit a revised version of the manuscript that addresses the points raised during the review process.

Please submit your revised manuscript within 60 days. If you will need more time than this to complete your revisions, please reply to this message or contact the journal office at digitalhealth@plos.org. Please include the following items when submitting your revised manuscript:

We look forward to receiving your revised manuscript.

Kind regards,

Hadi Ghasemi

Academic Editor

PLOS Digital Health

Hadi Ghasemi

Academic Editor

PLOS Digital Health

Journal Requirements:

Additional Editor Comments (if provided):

Reviewers' comments:

Reviewer's Responses to Questions

**Comments to the Author**

1. Does this manuscript meet PLOS Digital Health’s publication criteria? Is the manuscript technically sound, and do the data support the conclusions? The manuscript must describe methodologically and ethically rigorous research with conclusions that are appropriately drawn based on the data presented.

Reviewer #1: Yes

Reviewer #2: Yes

Reviewer #3: Partly

Reviewer #4: Yes

2. Has the statistical analysis been performed appropriately and rigorously?

Reviewer #1: Yes

Reviewer #2: Yes

Reviewer #3: I don't know

Reviewer #4: Yes

3. Have the authors made all data underlying the findings in their manuscript fully available (please refer to the Data Availability Statement at the start of the manuscript PDF file)?

Reviewer #1: Yes

Reviewer #2: Yes

Reviewer #3: No

Reviewer #4: Yes

4. Is the manuscript presented in an intelligible fashion and written in standard English?

Reviewer #1: Yes

Reviewer #2: Yes

Reviewer #3: Yes

Reviewer #4: Yes

5. Review Comments to the Author

Reviewer #1: This is an interesting, well-written paper. I think you bring up some important issues that 

will spark considerable debate, both in terms of what you have actually found and in terms of the policy implications. I think an interesting, but unexplored point is whether academics have changed their research strategy in light of the increased expectations of research production. One strategy in particular, using more co-authors is a way in which researchers can increase their productivity, even if space in top journals is limited. This strategy may be particularly salient to macro researchers, given the extremely limited space available.

Reviewer #2: This manuscript thoroughly addresses the issue of using natural language processing (NLP) to analyze text conversations between patients and healthcare providers during the COVID-19 pandemic. This paper shows how digital technologies, such as SMS, can be used to support isolated patients on a large scale and in conditions of health restrictions. Different machine learning models are thoroughly evaluated and a comprehensive comparison between the performance of the models in different languages is presented. The findings of the article emphasize that patient support is not limited to medical issues and social issues are also important. By providing solutions to improve health responses in future crises, this research has a high value for public health decisions.

Reviewer #3: • Usage of NLP within the context of healthcare services seems novel and interesting, but I couldn’t grasp whats the importance of conducting such study and how its findings can help the beneficiaries.

• Has the manuscript adhered to the journal`s guidelines in terms of format and sequence of sections? If it hasn’t adhered, I suggest the authors to revise the manuscript in accordance. 

• while the introduction is well-structured and informative, it could benefit from a more detailed discussion on the limitations of previous studies to highlight the novelty of their approach.

• Discuss the previous studies within the literature and the novelty of this study at the end of the introduction section. 

• Did u use any sampling technique for selection of the study sample? 

• it would be beneficial to include more information on the selection criteria for text messages analyzed and how potential biases were mitigated, particularly regarding language translation challenges with Kinyarwanda texts.

• Does the study have any ethical approval from the corresponding ethical committee? 

• What is the significance of the findings of this study? who can use such findings? Are these findings important for the beneficiaries? 

• The paper lacks discussion of results with the literature within the context. Elaboration on how these results compare with existing literature could strengthen their significance. 

• In summary, the methodology of the study and usage of NLP within such context seems interesting; Anyhow, I believe there is no clear rationale behind conduction of such study and I couldn’t grasp the research question fully. I suggest the authors to use the same methodology and utilize the same data once again, but this time with a clear research question which seems valuable for readers and beneficiaries. As an example, I believe the data within the text messages could be utilized to obtain an analysis on the elements and principles of patient-centered care within the study population; something which seems to be valuable for the beneficiaries within the context. OR analysis of the views of users of such SMS services provided within their text messages. OR an analysis on the challenges of service users during their utilization of SMS services. OR an analysis on the types of patients who had positive and negative experiences during usage of such SMS services which can provide significant implications for the beneficiaries. OR any other topic the authors believe as having the potential to provide valuable insights for the beneficiaries. With kind regards.

Reviewer #4: This study provides valuable insights into using SMS-based health monitoring for isolated patients during the SARS-CoV-2 pandemic in Rwanda. The large-scale analysis of patient conversations using NLP, especially in under-resourced languages like Kinyarwanda, is a significant contribution. The comparison of machine learning models further enhances the relevance of your findings for digital health. Your work demonstrates the scalability of remote support in pandemics and offers important insights for future public health responses.

6. PLOS authors have the option to publish the peer review history of their article (what does this mean?). If published, this will include your full peer review and any attached files.

**Do you want your identity to be public for this peer review?** For information about this choice, including consent withdrawal, please see our Privacy Policy.

Reviewer #1: No

Reviewer #2: Yes: Mohammad Ranjbar

Reviewer #3: Yes: Mohsen Khosravi

Reviewer #4: No

---

## [Editor Report · Decision Letter 1]

19 Nov 2024

Natural language processing to evaluate texting conversations between patients and healthcare providers during COVID-19 Home-Based Care in Rwanda at scale

PDIG-D-24-00364R1

Dear Richard Todd Lester

We are pleased to inform you that your manuscript 'Natural language processing to evaluate texting conversations between patients and healthcare providers during COVID-19 Home-Based Care in Rwanda at scale' has been provisionally accepted for publication in PLOS Digital Health.

Best regards,

Hadi Ghasemi

Academic Editor

PLOS Digital Health